# Indigenous Epistemological Frameworks and Evidence-Informed Approaches to Consciousness and Body Representations in Osteopathic Care: A Call for Academic Engagement

**DOI:** 10.3390/healthcare13060586

**Published:** 2025-03-07

**Authors:** Rafael Zegarra-Parodi, Thioro Loum, Giandomenico D’Alessandro, Francesca Baroni, René Zweedijk, Stéphan Schillinger, Josie Conte, Lewis Mehl-Madrona, Christian Lunghi

**Affiliations:** 1BMS Formation, 75116 Paris, France; francesca@bms-formation.com (F.B.); christian@bms-formation.com (C.L.); 2Institut d’Ethnologie, Université de Neuchâtel, 2000 Neuchâtel, Switzerland; thioro.loum@unine.ch; 3Clinical-Based Human Research Department, Foundation Centre for Osteopathic Medicine Collaboration, 65121 Pescara, Italy; gdalessandro@comecollaboration.org; 4Research Department, A.T. Still Academy Italia, 70124 Bari, Italy; 5Panta Rhei, 5121 ML Rijen, The Netherlands; rene@pro-osteo.com; 6Curieux Hasard, 67100 Strasbourg, France; steph.schillinger@gmail.com; 7University of New England College of Osteopathic Medicine, Biddeford, ME 04005, USA; josephine.conte@mainegeneral.org; 8Maine-Dartmouth Family Medicine Residency, Augusta, ME 04330, USA; 9Coyote Institute, Orono, ME 04473, USA; lewis.mehlmadrona@maine.edu

**Keywords:** body representation, cultural competence, consciousness, evidence-based medicine, Indigenous narratives, manual therapy, osteopathy, person-centered care, traditional healing

## Abstract

**Background/Objectives:** Indigenous perspectives, which emphasize non-materialistic dimensions of healing, such as the interconnectedness of the body, mind, and spirit, align with one foundational principle of osteopathic care. Integrating these perspectives into person-centered care may enhance therapeutic effectiveness by accommodating diverse understandings of health and well-being. This perspective paper explores how various epistemological frameworks, including Indigenous non-materialistic approaches, can inform manual therapy techniques and therapeutic alliances to advance person-centered care. **Methods:** We synthesized the best available evidence with expert insights and interdisciplinary viewpoints to address the gaps in the scientific literature. Our approach integrates conceptual analysis and emerging research to provide a comprehensive discussion for a broad professional audience. **Results:** We focused on detailing the existing sociocultural and experiential frameworks available to describe patients’ bodily perceptions rather than abstract intellectual constructs. Our findings were divided into two sections. The first examines the incorporation of diverse body representations that extend beyond purely biomechanical interpretations, emphasizing the role of non-materialistic components in therapeutic processes. The second explores recent neuroscientific research on self and consciousness, demonstrating how these insights intersect with Indigenous perspectives to enrich the theoretical and practical applications of osteopathic principles in different clinical contexts. **Conclusions:** Epistemological flexibility has the potential to refine clinical frameworks and ensure that they reflect the full scope of osteopathic practices beyond musculoskeletal care. By integrating diverse sociocultural perspectives without reinforcing stereotypes or rigid cultural constructs, this approach clarifies the diversity of body representations in osteopathic practices, addresses gaps in academic discourse, and promotes the integration of multiple worldviews as a foundation for truly person-centered care.

## 1. Introduction

Manual therapy professionals use similar hands-on techniques but interpret them through different theoretical frameworks, each shaped by their own historical and conceptual background. In their early writings, these professions emphasized unique concepts and specialized manual skills to establish legitimacy and authority in healthcare [1]. This diversity of professions has created narratives that align with patients’ values and expectations, supporting individualized, person-centered care models. However, it has also contributed to inter- and intra-professional conflicts, particularly regarding the use of early vitalistic frameworks, which are increasingly challenged by Western biomedical paradigms [2,3]. Vitalism, rooted in both Western and non-Western traditions, has profoundly shaped health perspectives in Western thought, since ancient Greece. The following two prominent schools of medical philosophy emerged in Greece: the Aesculapian approach, which adopted a mechanistic view of health and attributed illness to material causes, and the Hygeian approach, which emphasized holistic, vitalistic principles. The latter recognized the body’s intrinsic self-healing abilities and defined healthcare providers as guides or facilitators in supporting these natural processes. Today, this dichotomy persists, with biomedicine reflecting the Aesculapian model, while integrative health professions and traditional healing practices are more aligned with the Hygeian philosophy [4]. In contrast, the holistic approach emphasizes the interconnectedness of the body, mind, and spirit, rather than focusing solely on symptoms. Medical practice highlights the role of biological, psychological, social, and spiritual factors in shaping health and illness [5]. Emerging evidence provides healthcare practitioners and academics with an opportunity to reassess their belief systems and evaluate their impact on patient care, which could also apply to theories of consciousness, representing a critical yet often overlooked aspect of the body–mind–spirit paradigm in clinical care. Materialist theories of consciousness typically adopt a physicalist stance characterized by naturalistic and science-based approaches. In contrast, nonmaterialist theories encompass a variety of non-physicalist views that often extend beyond current scientific understanding and, in some instances, are not amenable to empirical experimentation or replicability [6]. Nonmaterialist theories, which incorporate spiritual and existential dimensions to better understand worldviews, may challenge conventional Western academics and epistemologies, while remaining fundamental to Indigenous healing traditions, potentially offering valuable insights into contemporary manual therapy practices [7]. In the United States, osteopathic care is a licensed medical profession in which doctors of osteopathic medicine (DO) are trained in both conventional medicine and osteopathic principles, combining medical care with osteopathic manipulative treatment (OMT) to address physical symptoms and promote overall health. Outside the U.S., osteopathy is primarily practiced as a manual therapy discipline that focuses on musculoskeletal health and wellbeing [8]. Osteopathy was originally introduced at the end of the 19th century as an alternative to conventional medical care in the context of frontier America, emphasizing hands-on treatments within a nonmaterialist framework. As Dr. A.T. Still, the founder of osteopathic medicine, stated: “God manifests himself in matter, motion, and mind. Study well his manifestations” (Figure 1). Today, osteopathic principles continue to emphasize the integration of body, mind, and spirit, drawing on a holistic approach inspired by Indigenous Native American traditions. These traditions, including the medicine wheel framework, highlight balance and harmony across the physical, mental, spiritual, and emotional dimensions [9,10]. Recent studies on touch-based therapies within a holistic framework have shown benefits for physical and psychological health, emphasizing the potential of osteopathy to integrate diverse paradigms beyond Western materialist models [11].

Despite the cultural and historical legacies that have shaped broader conceptions of health within the osteopathic profession, integrating nonmaterialistic perspectives into osteopathic education and practice remains a challenge [12,13]. This highlights the need to enhance person-centered care by incorporating diverse worldviews on body representation in health and disease, extending beyond Western biomedical models. New evidence on the role of bodily awareness in fostering experiences of unity and interconnectedness suggests that insights derived from millennia of observation within Indigenous healing traditions may offer valuable perspectives for integrating the body, mind, and spirit in therapeutic practices, such as osteopathic care [14]. Therefore, integrating neuroscience, particularly the neuroscience of the self, with Indigenous wisdom may provide a holistic embodied approach to understanding human health, particularly within manual therapy. Supporting this, a systematic review of touch-based therapies identified 45 clinical studies that framed interventions holistically, showing that manual therapies improve overall wellbeing, alleviate symptoms, and promote relaxation, while ensuring safety [11]. Tyreman’s anthropological–ecological narrative highlights key osteopathic principles, viewing humans as interconnected organisms rather than mechanical systems and prioritizing the individual over the disease. This perspective enriches the recognition of diverse body representations in osteopathic care, highlighting the dynamic interplay between individuals and their sociocultural environments that shapes these representations [15]. The epistemological flexibility inherent in osteopathic care enables the integration of diverse paradigms beyond Western materialism, including nonmaterialist frameworks found in Indigenous traditions. Such knowledge transfer could expand manual therapy by incorporating the non-physical aspects of health, such as the influence of patients’ beliefs on consciousness and body representation, which have been historically explored in philosophy and religion but are now being studied by neuroscience, potentially facilitating the integration of these insights into Western health paradigms. Integrating these concepts into education and clinical practice remains challenging, as some osteopathic academics increasingly dismiss approaches beyond the conventional musculoskeletal care paradigm [12,13].

This perspective paper aims to address the gap in osteopathic care by integrating diverse worldviews beyond materialistic frameworks with a focus on body representations in clinical practice. It examines how various epistemological frameworks, including Indigenous nonmaterialistic perspectives, could inform manual skills and therapeutic alliances to advance person-centered care and inform existing clinical paradigms. Indigenous perspectives, emphasizing nonmaterialistic dimensions of healing, such as the interconnectedness of the body, mind, and spirit, align with the foundational principles of osteopathy and may enrich person-centered care by addressing diverse understandings of health and wellbeing [16]. However, due to the current lack of robust evidence supporting these frameworks, preliminary work is necessary to define key concepts. The primary research question is as follows: how could Indigenous epistemological frameworks rooted in nonmaterialistic views on consciousness contribute to the development of osteopathic practice, which is currently constrained by materialistic paradigms?

## 2. Methods

This perspective paper was developed in accordance with established guidelines for writing commentaries, a methodological approach commonly employed to initiate new areas of investigation when data are lacking [17]. To address the objectives of this paper, the authors examined relevant data on consciousness and body representation to bridge the academic gap between materialistic and nonmaterialistic perspectives within manual therapy. The theoretical framework for this commentary was crafted by a team of experts (WW, XX, YY, and ZZ) with over 10,000 h of professional experience in education, scientific research, and clinical osteopathic practice. This framework emerged from a collaborative brainstorming process grounded in clinical observations and the best available evidence. To ensure methodological rigor, a quality assessment tool for narrative review articles was used. A comprehensive literature search was conducted from September to October 2024 using MEDLINE (PubMed), EMBASE, and Google Scholar. Search terms (e.g., body representation, cultural competence, consciousness, evidence-based medicine, Indigenous narratives, manual therapy, osteopathy, person-centered care, and traditional healing) were tailored to each database, and relevant subheadings were applied, where appropriate. The search was restricted to articles published in English, with no limitations regarding the study design, population, outcomes, or publication date. The reference lists of the identified articles were also reviewed, and a snowball sampling approach was employed to uncover additional relevant studies. No formal validity or quality assessments were performed on the selected articles to capture the full spectrum of available information on the topic. The selection process involved two stages and was conducted independently by the authors (WW, XX, YY, and ZZ). First, the abstracts of eligible articles were reviewed to determine their relevance to the commentary. Subsequently, full-text versions of the selected studies were screened using the same criteria.

## 3. Results

To explore the potential contributions of Indigenous epistemological frameworks, which are grounded in nonmaterialistic perspectives on consciousness, to the advancement of osteopathic practice—often constrained by predominantly materialistic paradigms—we structured our findings into two key sections. The first section, Body Representations in Manual Therapy, examines the integration of diverse body representations that transcend purely physical interpretations, emphasizing the role of nonmaterialistic components in therapeutic processes. The second section, Insights from the Neuroscience of the Self: Integrating Non-Physical Body Dimensions and Cross-Cultural Influences in Osteopathic Care, explores recent neuroscientific findings on the self and consciousness, highlighting how these insights intersect with Indigenous perspectives to enrich the understanding and application of osteopathic principles and practices across different clinical contexts.

### 3.1. Body Representations in Manual Therapy

#### 3.1.1. Integrating Physical and Non-Physical Dimensions into Body Representations

Understanding body representations enhances our ability to describe and observe human experience. These representations are frameworks through which societies structure their understanding of the body, not just as a biological entity but also in relation to social interactions and the environment. They shape how individuals perceive, engage with, and interpret the world, guide social interactions, and provide context for daily life [18]. The body serves as a bridge between inner experiences and the external world, actively mediating our understanding of both. Rather than being passive, the body is an active site of perception that is central to human experience, as emphasized by phenomenology [19]. Perception and consciousness are inherently embodied, with the body as the primary medium for interpreting the world [20]. Embodiment raises significant questions regarding the relationship between consciousness and neural processes, such as how mental states (e.g., sensory, cognitive, and emotional) relate to brain states. The connection between phenomenal consciousness, cognition, and brain state has been widely debated in philosophy, neuroscience, and psychology. Kuhn [6] recently reviewed 209 theories of consciousness, categorizing them into 10 major classifications. Key theoretical positions include epiphenomenalism and material monism, which suggest that thoughts and feelings are secondary to brain processes and do not influence physical behavior. In contrast, dualist and idealist perspectives argue that consciousness cannot be fully explained by brain activity, emphasizing its non-physical nature [20]. The body encompasses both physical and non-physical components, with non-physical aspects, such as consciousness, shaped by social and cultural influences. This ongoing debate highlights the complexity of understanding consciousness and its intricate relationship with brain function, emphasizing the need for further interdisciplinary investigation [6,21]. Drawing from cross-cultural research and medical anthropology, Kleinman developed a model to address the limitations of conventional biomedical approaches, which often prioritize physical factors, while neglecting broader considerations. The model outlines three key dimensions: illness, the subjective experience of being unwell, which includes personal perceptions, emotions, and meanings; disease, the biomedical perspective focused on diagnosing and treating physical dysfunction; and sickness, which encompasses the social and cultural dimensions of illness, including its effects on societal roles and interpersonal relationships [22]. This perspective offers a framework to facilitate the integration of the non-physical dimensions of the body, such as the mind, spirit, and emotions, into Western biomedical paradigms. It conceptualizes the body as both a biological entity and vessel shaped by cultural and social practices. This highlights how societies teach posture, movement, self-care, and other embodied behaviors, reflecting and internalizing social norms at a young age [23]. By incorporating the diverse “social” dimensions of the biopsychosocial model, this approach improves its application in musculoskeletal care, promoting a more comprehensive understanding of patient needs that includes cultural, social, and psychological factors alongside biological aspects [7].

#### 3.1.2. Western Biomedical Construction of Mind–Body Duality

In Western societies, mind–body duality arises from social structures and tensions between collective and individual identities. Dorian and Garfinkel noted that societal expectations often pressure individuals to exceed their natural limits, leading to conflicts between bodily representations and societal demands [23]. This duality is deeply embedded in biomedical logic and shaped by technological advances, scientific paradigms, and transhumanist ideals, which objectify the body as a technical entity subject to manipulation and control. For instance, practices such as organ transplantation raise anthropological concerns about body fragmentation, challenging the notion that the body is integral to one’s identity. Furthermore, a pervasive societal obsession with youth, health, and performance perpetuates the authority of biomedicine, reinforcing a Cartesian framework centered on logic and methodical reasoning [23]. While advancing medical knowledge, this approach limits holistic perspectives on health, with biomedicine’s dominance closely tied to societal values that prioritize efficiency, rapid progress, and immediate results [24].

A hierarchical structure that positioned the biomedical model as dominant was established using the Flexner Report, marking the beginning of the marginalization of holistic and Indigenous healing systems [25]. Furthermore, mind–body duality varies within the same culture and is shaped by differing social structures that influence common social models. Scheper-Hugues and Lock [26] introduced the concept of a three-dimensional body that challenges traditional views of the body by examining three key perspectives: (1) the phenomenologically experienced individual body self; (2) the social body, symbolizing relationships among nature, society, and culture; and (3) the body politically influenced by social and political control. The significance of local contexts on biological factors in shaping global health policies is crucial, as they recognize the social and cultural dimensions of health as integral to individual wellbeing. By contrast, viewing the body as a machine, detached from its sociocultural context, offers a limited perspective that fails to capture the complexity of human health and experience [23]. The dematerialization paradigm, which advocates for a “farewell to the body” [27], signals a shift toward a post-biological, post-organic, and almost transhumanist reality, challenging traditional representations of holistic and integrative medical care. This paradigm shift introduces a separation between individuals and their bodies, detaching them from the interconnectedness of the cosmos, community, and the self. Consequently, the body is perceived as a mechanical entity that is an accessory to an individual that can be manipulated, repaired, or transformed. In response to the Cartesian framework in biomedicine, which limits integrative health approaches, psychosomatic medicine has introduced the biopsychosocial model, incorporating psychological and social factors into the biological understanding of health [3]. Despite these advancements, Engel [28] highlighted the dominance of biomedical models, which continue to shape clinical practice. He also emphasized that incorporating psychosocial aspects into healthcare requires emotional and moral engagement from clinicians, presenting a challenge to the established framework. This tension is particularly evident when symptoms lack a clear biological basis and are instead attributed to psychosocial factors. This objectified view of the body contrasts considerably with the inclusive and integrative principles essential to a holistic approach to health, often creating tension with the prevailing biomedical models. The biopsychosocial model, while designed for Western contexts, shares similarities with Indigenous healing practices in its recognition of the non-physical dimensions of health [7]. This model can enhance therapeutic responsiveness by moving beyond standardized frameworks to better address diverse patient beliefs and health needs, whether culturally transmitted or shaped by life experiences, as these may evolve over time for each individual [7]. Incorporating decolonial perspectives within a global framework could improve healthcare by acknowledging the broader economic, social, and political contexts that influence health. Unlike Western biomedical models, which often separate the body and mind, many Indigenous cultures view the body as an interconnected whole, integrating the physical, spiritual, and emotional dimensions [10]. Exploring the intersection of cultural narratives, body representations, and Western frameworks could foster more holistic and inclusive care models.

#### 3.1.3. Indigenous Healing Traditions: A Lived Bodily Experience Beyond Intellectual Processes

Indigenous peoples are defined as communities with historical connections to precolonial societies, maintaining distinct cultural, social, and economic practices within their ancestral territories. They constitute approximately 5% of the global population, or approximately 370 to 500 million individuals, yet they represent a greater part of the world’s cultural diversity. Indigenous peoples occupy, own, or use 22% of the global land area, and speak 7000 languages, contributing significantly to linguistic diversity. These communities are spread across 70 countries, emphasizing the importance of self-identification and preservation of their cultural and territorial rights, as outlined in key documents, such as the International Labor Organization Convention No. 169, the United Nations Declaration on the Rights of Indigenous Peoples, and the United Nations Educational, Scientific, and Cultural Organization (UNESCO) conventions [29]. Indigenous peoples are essential for preserving cultural and biological diversity, yet they face ongoing marginalization, poverty, and human rights violations. The emergence of Indigenous research methodologies addresses the violence enacted through healthcare and research institutions and positions research as a process of cultural reclamation. To that end, frameworks such as the Coin Model of Privilege and Critical Allyship, which highlight the advantages and disadvantages of social structures, have been introduced to better understand the integration of Indigenous worldviews into healthcare. While these frameworks challenge the Eurocentric foundations of science and healthcare, they offer a transformative approach that contributes to reducing health inequities and addressing the social determinants of healthcare [30].

In many Indigenous cultures, the body is viewed not just as a biological entity but as a connection between the individual and the cosmos. The Indigenous knowledge of the body is deeply connected to a worldview in which it exists in a constant relationship with natural forces, spirits, and cycles of life. Lock’s concept of “local biologies” highlights that perceptions of the body vary significantly across cultural contexts, profoundly impacting healthcare practices [26]. For Indigenous populations, the body serves as a meeting point between the human and non-human realms, a space where intricate relationships with the environment and the sacred are expressed. Thus, it is regarded not just as a biomedical object but also as a living and meaningful entity with significant implications for health and healing practices. Physical symptoms are often interpreted as manifestations of an imbalance that may extend beyond their biological origin to include various aspects of the human experience. Indigenous healing practices usually involve bodily experiences through rituals that connect individuals to a universal consciousness and explore personal and collective identities [10]. To aid the understanding of readers unfamiliar with Indigenous traditions and rituals, three levels are introduced to illustrate the progression from the individual’s inner world to their interactions within groups and, ultimately, to their connection with universal or transcendent dimensions—elements often overlooked in Western biomedical materialistic worldviews but essential to Indigenous perspectives (Table 1).

The main challenge for Western academics regarding nonmaterialistic views of consciousness is the lack of reproducible empirical evidence that can validate these perspectives within mainstream scientific paradigms. However, Indigenous nonmaterialistic worldviews provide valuable insights into complex subjective experiences, such as near-death phenomena, which share features with Indigenous rituals involving Amazonian entheogenic plants and are now amenable to scientific investigation [31]. Indigenous healing traditions often integrate ritualistic components designed to facilitate the transition from ordinary to non-ordinary states of consciousness. These practices, observed globally, include the use of medicinal plants (such as entheogens), rhythmic drumming, repetitive chanting, physical exertion, fasting, and dancing [32]. The resulting physical manifestations described within nonmaterialistic frameworks of consciousness are not viewed as the primary objective but rather are systematically observed and described as integral elements of the psychointegrative process for therapeutic purposes [32]. These altered states of consciousness, which may include ego dissolution and mystical experiences, provide valuable examples of how Indigenous epistemological frameworks can enhance our understanding of biologically driven human experiences [33]. Indigenous healing perspectives introduce a spiritual and existential dimension closely linked to physical lived experiences, which are often absent in Western paradigms. These epistemological frameworks broaden our understanding of body representations and the diversity of embodied experiences, offering an opportunity to enrich integrative healthcare frameworks including manual therapy.

Recognizing the interconnectedness of the body with both physical and spiritual realms, Indigenous worldviews provide valuable insights that could enrich the understanding of health and healing in Western practices [11]. By embracing these interconnected dimensions, healthcare providers can cultivate a more holistic understanding of health that transcends the constraints of biopsychosocial models adapted for manual therapy, which frequently adhere to philosophically mechanistic and culturally dichotomous worldviews. This holistic perspective aligns with core osteopathic principles that prioritize treating the whole person, rather than focusing solely on isolated symptoms, and highlights the significance of spiritual and existential dimensions in healthcare, in accordance with the osteopathic tenet of considering each person as a dynamic interaction of body, mind, and spirit [34]. Exploring the Indigenous legacy of the body–mind–spirit osteopathic tenet may provide a clearer understanding of the goal of osteopathic care: to restore harmony by engaging with patients’ lived bodily experiences rather than relying solely on intellectual understanding or passive techniques. Integrating the physical, emotional, and existential dimensions into clinical practice could enhance treatment outcomes and promote a more holistic view of health beyond musculoskeletal symptoms.

#### 3.1.4. Reintegrating Indigenous Perspectives: The (R)evolution of Osteopathy Toward a Holistic Understanding of Body and Health

A recent qualitative study combining ethnographic observations of osteopathic treatment sessions and interviews with practicing osteopaths examined how practitioners perceive the body in clinical care [35]. While the authors contrasted the biomedical view of the body as a universal material entity with a more culturally contingent perspective, they did not formally explore the non-physical aspects of the body, such as those central to Indigenous traditions, despite referencing the body–mind–spirit osteopathic tenet [35]. This concept reflects an Indigenous legacy in which health is viewed as a state of harmony, with the body being an integral part of a broader interconnected system encompassing nature and the cosmos, rather than as an isolated entity [36]. Comparable Indigenous perspectives on body representations are observed across diverse cultures worldwide, highlighting the universal relevance and applicability of these concepts beyond the specific historical and geographical context of the origins of osteopathy in the U.S. Specifically, Native American cosmologies view the body as influenced by the same elemental forces—earth (stability), water (adaptability), fire (vitality), and air (spirit)— which shape the environment. This holistic framework, exemplified by the medicine wheel, integrates context, mind, body, and spirit, thereby offering a comprehensive model of wellbeing [10]. Unlike Western biomedical models, which often isolate symptoms, Indigenous approaches view illness as an imbalance between relationships with others, the environment, and the spiritual realm. Osteopathic care embodies this perspective by integrating existential and spiritual dimensions into healthcare and fostering epistemological flexibility in manual therapy. This approach incorporates diverse knowledge systems of body representations into clinical practice, aligning more closely with patients’ values and expectations [7]. This flexibility acknowledges the historical connections between Dr. A.T. Still and Native American populations but also facilitates the inclusion of Indigenous body representations. Such narratives emphasize the importance of culturally sensitive, person-centered care that incorporates patients’ sociocultural health beliefs shaped by life experiences, while avoiding cultural stereotypes [7]. This holistic perspective is central to osteopathic care and differentiates it from other manual therapy professions, facilitating the integration of nonmaterialist theories of consciousness and their influence on body representations. This approach provides valuable insights that can complement and expand biologically driven Western frameworks, enhancing our understanding of the non-physical dimensions of health. However, for this integration to be truly effective, practitioners must recognize and acknowledge the potential influences of materialistic versus nonmaterialistic worldviews on the dynamic interactions of the spiritual, emotional, and physical components of each individual [6]. Indigenous healing traditions demonstrate how body representations can shape patients’ values and expectations. Key practices, such as touch, mobilization, manipulation, and attentive listening, aim to restore balance within the body’s components [30]. These healing practices extend beyond a purely anatomical perspective, offering a comprehensive view of the individual. In this context, they conceptualized the body as a vessel for lived experiences, where unresolved emotions may manifest as physical pain, often referred to as psychosomatic in biomedical contexts [36].

Building on this understanding, the body–mind–spirit osteopathic tenet transforms the body into language and a series of messages to be deciphered, with the practitioner acting as an attentive interpreter, helping the patient release the body from invisible non-physical imprints. Palpatory signs associated with somatic dysfunction (e.g., changes in stiffness, elasticity, tension, and mobility restriction) can reflect an individual’s history of trauma, emotions [37,38], and accumulated stress and allostatic load [39]. Contemporary osteopathic care highlights the interconnectedness of the human body, in which the functionality or dysfunctionality of each anatomical structure is influenced by several factors, including (1) the condition of other structures within the body; (2) the self and its relationship to materialistic versus nonmaterialistic worldviews on consciousness, encompassing the individual’s subjective experiences, awareness, and efforts; (3) culture and shared values that reflect intersubjective interactions; and (4) social systems and structures, including environmental factors and broader social contexts [38]. Similar to Indigenous traditions, this approach empowers patients by reframing their perception of the body from a purely anatomical entity to a complex system that is integrated with emotional, psychological, and spiritual dimensions. Practitioners facilitate opportunities for patients to reflect on their bodily histories, promoting awareness of the interplay between emotions and physical symptoms, consistent with the current osteopathic principles [34]. Engaging in these narratives enhances practitioners’ understanding of patients’ cultural background and health experiences. However, a theoretical framework that supports this holistic approach is currently lacking.

From an epistemological perspective, Gélinas and Bouchard [40] recognized that Indigenous and scientific knowledge systems are fundamentally different ways of understanding the world. They proposed a framework to bridge this gap, focusing on the following two key principles: maintaining neutrality regarding different knowledge types and clearly defining how knowledge can be transferred between them. However, such an approach requires a detailed definition of what constitutes knowledge in both systems, to ensure effective communication and integration. The challenge is that if only scientific criteria are applied, Indigenous knowledge may be dismissed as a simple belief [40]. Recent advancements in the neuroscience of consciousness and the bodily self have brought complex human experiences, central to Indigenous knowledge and traditionally explored through philosophy and religion in the Western context, into the medical field [33]. This potential for knowledge transfer could facilitate the integration of these insights into fields, such as osteopathic education and practice. By doing so, it could support the inclusion of non-physical dimensions of the body in clinical care, while considering their relationships with cross-cultural influences and placebo effects. Embracing such a framework could enhance patient–practitioner relationships, enrich the therapeutic process, and ultimately, improve clinical outcomes in osteopathic care.

### 3.2. Insights from the Neuroscience of the Self: Integrating Non-Physical Body Dimensions and Cross-Cultural Influences in Osteopathic Care

In Western healthcare, particularly in manual therapy, spiritual experiences are often excluded from academic and scientific discourse, despite their relevance to patient care [3]. This omission overlooks the potential influence of patients’ beliefs about consciousness and their impact on body representations in health and disease, which may affect the outcomes of touch-based treatments [11]. Research in the neuroscience of consciousness has made it possible to empirically explore the connection between brain activity and human experience [33]. This emerging field bridges the gaps between science, philosophy, and holistic healthcare by integrating the spiritual and existential dimensions. The following two primary frameworks explore consciousness: materialist theories, which view it as an emergent property of neural processes; and nonmaterialist theories, which suggest that the brain may receive consciousness from a broader field [6]. The latter challenges traditional paradigms and encourages the study of mystical and existential experiences that have been previously neglected. These frameworks influence body representation and symptom interpretation, offering opportunities for personalized and belief-aligned care [12]. The predictive processing (PP) theory, which models the brain as a Bayesian system that integrates and updates sensory information, further supports this understanding. PP explains how multisensory integration supports body awareness and decision making [41]. Adopting this framework could enhance osteopathic care by addressing body-based experiences and integrating spiritual components into holistic patient-centered approaches [7]. Specifically, the concept of multisensory integration, in which data from various senses are combined to create a coherent perception of oneself and the surrounding environment, has significant implications for understanding spiritual and existential experiences. Changes in multisensory integration can influence patients’ body perception, often observed in altered states of consciousness, such as mystical experiences or out-of-body phenomena. Recent research indicates that these experiences may be linked to variations in how interoceptive and exteroceptive signals are weighted, alterations in sensory error monitoring, or individual differences in brain function and structures associated with interoception [33]. These findings suggest that the spiritual/existential dimension, often regarded as metaphysical, can be empirically assessed, providing valuable insights for osteopathic practitioners treating patients with complex symptoms. This framework aligns with the body–mind–spirit osteopathic tenet and facilitates the integration of nonmaterialistic perspectives on consciousness, broadening the scope of osteopathic care beyond a musculoskeletal focus [7]. This holistic approach, accessible to all patients and not just Indigenous individuals, allows touch-based therapies to go beyond the physical application of techniques and engage with the whole person, including the existential and spiritual dimensions of wellbeing [12]. Embracing this epistemological flexibility allows osteopathic care to be both evidence-informed and ethically sensitive, respecting the unique values and beliefs of each patient that can evolve over time [7].

Pain neuroscience in musculoskeletal management highlights the role of contextual factors in specific manual effects, emphasizing the importance of enhancing placebo effects and minimizing nocebo effects [42]. Anthropological perspectives have expanded our understanding of the placebo effect by recognizing its cultural and psychological dimensions, moving beyond a purely physiological view [43]. This challenges the reductionist biomedical approach to manual therapy, advocating for an integrative healthcare model that incorporates psychological, social, and cultural factors into the treatment of illness and promotion of healing. Indigenous narratives may enrich patient-centered care by respecting and incorporating patients’ cultural beliefs, values, and perspectives, while embracing nonmaterialistic views on consciousness and body representation central to Indigenous frameworks [6]. This approach, grounded in the universal principles of care, transcends ethnically specific applications. Native American perspectives significantly influenced Dr. Andrew Taylor Still, whose 19th century holistic practices continue to shape integrative healthcare [14]. Recognizing diverse patient values and expectations, particularly body representations and preferred therapeutic approaches, is crucial for providing effective care. Groenevelt and Slatman [35] suggested that osteopathic touch facilitates body perception by (1) refining tactile sensitivity, (2) understanding physiological processes, (3) attuning to the sensory properties of tissues, and (4) interpreting the body’s signals, including the associated emotional components, beyond its biological boundaries. Additionally, Baroni et al. [44] explored the distinctive aspects of osteopathic touch in clinical practice, highlighting how palpatory cues of somatic dysfunction can be integrated into a participatory patient-centered model of osteopathic care. This diversity highlights the socially mediated placebo effect, showing how healthcare providers’ behaviors and cognitive models impact clinical interactions [45]. Practitioners with greater epistemological flexibility regarding body representations in manual therapy can offer a broader spectrum of treatment options than those strictly adhering to a single theoretical framework, whether materialistic or nonmaterialistic [7]. However, recognizing Indigenous knowledge in health necessitates a reassessment of marginalized epistemologies that are often dismissed by modern science [24,46]. By recognizing the value of Indigenous bodily knowledge, osteopathic care can both enrich its scope of practice and could contribute to redefine what it means to “care”. The challenge here is twofold: First, recognizing that Indigenous knowledge is a living, adaptive knowledge that has much to offer in a world characterized by ecological and health crises. In contrast, the osteopathic profession needs to better explore this knowledge by engaging in a genuine dialogue that respects the depth of this knowledge in contemporary osteopathic practice. To this end, an osteopathic model of patient–practitioner–environment synchronization aligned with person-centered care principles and evidence-based frameworks informed by Indigenous epistemological frameworks has been proposed [7]. This model, which is applicable in contemporary settings, fosters patient–practitioner synchronization through interprofessional aspects, such as effective communication, patient involvement, and non-verbal approaches, such as touch and proximity [47].

## 4. Discussion

Osteopathic care offers a unique opportunity to integrate contemporary scientific frameworks with osteopathic principles rooted in both Western and Indigenous traditions, fostering a more holistic approach. However, such integration presents several challenges. To support this transition and encourage academic engagement, educators and clinicians need information and guidance on how various theories of consciousness influence body representations in health and disease. Developing culturally sensitive methodologies that embrace epistemological flexibility, derived from the original dual foundation of osteopathy, is essential for distinguishing osteopathic care from other forms of manual therapy and for enhancing person-centered care. To support the integration of emerging fields, such as consciousness theories, into clinical care, we recommend that academics and clinicians follow the guidance of public organizations, such as the National Center for Complementary and Integrative Health (NCCIH) in the U.S. The NCCIH focuses on providing scientific evidence regarding the safety, effectiveness, and potential benefits of therapies that fall outside of conventional medicine, helping to inform healthcare decisions and policies. In relation to this paper, the NCCIH has offered valuable guidance to advance the knowledge of the neural basis of mind–body pain therapies [48]. Recently, a non-pharmacological intervention (NPI) model, specifically applicable to manual therapy, was introduced in France [49]. This model provides 14 ethical guidelines and 63 methodological recommendations, categorized into the following five types of NPI evaluation studies: (1) explanatory mechanisms and processes (mechanistic), (2) the content of practices (prototypical), (3) the evolution of practices (observational), (4) the benefits and risks of the NPI (intervention), and (5) strategies for application and personalization (intervention). The adoption of such a standardized evaluation framework for osteopathic care, extending beyond the traditional musculoskeletal scope, has the potential to benefit patients, practitioners, and third parties. This supports the promotion of evidence-informed approaches that accommodate diverse body representations, including those rooted in nonmaterialistic worldviews on consciousness.

An example of the potential usefulness of Indigenous epistemological frameworks in osteopathic practice is found in Indigenous healing practices, particularly those that involve “laying-on-hands” techniques, described across various traditions. These practices often target body regions that are not anatomically related to the pain or symptoms being treated [44]. This approach can be understood as an interoceptive manual technique used by osteopathic practitioners, characterized by gentle, prolonged light touch without joint mobilization. In a randomized placebo-controlled trial, such touch-based therapy has been reported to influence brain function related to interoception in patients with low back pain in a randomized placebo-controlled trial [50]. Furthermore, a rigorously conducted trial demonstrated that different forms of practitioner attention during an interoceptive manual approach, involving 15 min of static touch, while the practitioner focused on either a tactile or auditory attention task, significantly influenced brain functional connectivity in healthy individuals [51]. Although this may challenge materialistic paradigms, integrating a nonmaterialistic framework with specific narratives could provide a more comprehensive understanding of therapeutic touch, as traditionally applied in osteopathic techniques, such as visceral, cranial, and biodynamic approaches. Similarly, Indigenous epistemological frameworks can inform the therapeutic alliance in osteopathic care by integrating non-physical dimensions, such as existential and spiritual aspects of healing. These dimensions are already recognized in the clinical guidelines for end-of-life care, where they help provide meaning and purpose for patients. This concept was recently introduced in an osteopathic biopsychosocial–spiritual care model [52]. This biopsychosociospiritual construct was further explored by Fahlgen et al. [53], who surveyed 524 participants to investigate how health personality is shaped by the interactions between temperament (body), character (mind and soul), and external factors (social influence). The authors used the biopsychosocial model of personality developed by Cloninger to acknowledge the interplay between the body, mind, and soul in osteopathic care. They observed that self-transcendence was positively associated with life harmony, resilience, positive affect, and energy. However, nonmaterialistic perspectives grounded in bodily experiences have not been explicitly addressed, and discussions remain within materialistic Western frameworks [53]. A pilot study evaluated the impact of a 3-day body–mind–spirit intervention program designed to foster a holistic approach to health and wellbeing in 44 participants [54]. This program aimed to raise awareness, develop strength, and discover meaning in daily life. Post-intervention, participants reported significantly lower levels of physical distress and negative affect, along with increased positive affect, spiritual resilience, tranquility, and daily functioning [54]. These results suggest that integrating holistic practices into professional education could enhance both practitioner wellbeing and clinical outcomes by adopting a similar strength-based approach based on the belief that individuals possess the inner resources necessary to navigate life’s challenges effectively.

A roadmap was proposed to evaluate the clinical relevance of the body–mind–spirit osteopathic tenet in contemporary care [55]. This roadmap highlights the potential integration of this tenet into clinical practice to address the professional skills increasingly challenged by the current evidence. This roadmap also expands on the Cynefin framework, which was introduced in osteopathic care to guide clinical decision making. It helps practitioners manage clinical complexity and is based on clinical observations from the full diversity of OMT described in the professional literature [56]. This framework helps practitioners create narratives and approaches that align with patients’ diverse health beliefs, improving their understanding of body representations in both symptomatic and asymptomatic osteopathic clinical scenarios [14]. The Cynefin framework illustrates the complexity of a patient’s history, enabling both patients and practitioners to maintain a holistic perspective that recognizes the impact of psychological and existential domains on health, while integrating biological and biomedical factors. Additionally, it supports shared clinical decision making and the development of culturally sensitive, person-centered osteopathic care by providing flexibility to address both materialist and nonmaterialist worldviews [7] (Figure 2).

Academics should be encouraged to develop clinical frameworks that reflect the full diversity of current osteopathic practices, while avoiding extreme positions, such as pseudoscience and scientism, in the misuse of scientific methods in healthcare (Table 2). This paper emphasizes the proper application of science, grounded in critical thinking, to advance evidence-informed practices and challenges the academic tendency to limit osteopathic care to the boundaries of Western-based frameworks commonly used by other manual therapy professionals. We argue that the uniqueness of osteopathic principles lies in their integration of Western health paradigms with the body–mind–spirit tenet, which is drawn from Indigenous cultural perspectives. This integration allows for the inclusion of nonmaterialistic worldviews that challenge the constraints of Western epistemologies, which often restrict care to musculoskeletal practices and do not reflect what patients historically and uniquely sought in osteopathic care. Before implementing such a roadmap, it is essential to examine the intellectual humility process. This involves acknowledging the importance of new evidence and being open to revising the existing belief systems. Practitioners and academics must recognize that knowledge is constantly evolving and that the integration of diverse perspectives, particularly those that challenge conventional paradigms, can lead to more effective and inclusive approaches. By embracing this mindset, one can better navigate complex issues [7,56], foster critical thinking, and ensure that practices remain adaptable and responsive to emerging insights (Table 2).

## 5. Conclusions

In conclusion, this perspective paper highlights the critical role of recognizing diverse body representations in osteopathic care. These representations, which reflect how individuals perceive and interpret sensory information from their bodies, are influenced by factors such as culture, identity, and social context and may evolve over time. These influences profoundly affect healthcare experiences, shaping the perception of symptoms, illness, and disease, as well as decisions regarding when and how to seek care. The body is central to the construction and expression of identity, with gender, race, and social hierarchies affecting how bodies are perceived, treated, and understood within different healthcare systems. This integration has the potential to go beyond one-sided views and bridge the historical divide between science, philosophy, and holistic healthcare by incorporating new insights into the diversity of consciousness theories and their influence on body representations in health and disease. Incorporating insights from the long-standing traditions of nonmaterialistic worldviews, such as Indigenous narratives and body representations, into contemporary practice enriches our understanding of health and improves patient care by fostering inclusive and culturally sensitive approaches. Osteopathic care as a holistic discipline may benefit from moving from ethnocentric viewpoints to embrace the lived experiences, narratives, and cultural contexts of patients, thereby embracing the richness of diverse worldviews. This shift allows practitioners to treat patients by addressing both the physical and non-physical dimensions of the body. A key challenge lies in engaging with the diverse knowledge systems of patients, while placing the body at the center of care and respecting its subjective, social, and cultural contexts. Therefore, drawing from the observation of non-Western epistemologies regarding non-physical components of the body, such as Indigenous representations, is not just about adding cultural knowledge but necessitates a fundamental transformation in how academics and practitioners approach health, illness, and disease. This transformation requires ongoing education, openness to diverse worldviews, and a critical understanding of the limitations of current biopsychosocial models, particularly for patients with nonmaterialistic perspectives. As osteopathic care evolves, integrating theories of consciousness and addressing the shortcomings of traditional biomedical paradigms can open new pathways for enhancing biobehavioral synchronization with patients’ environments and diverse social contexts. Intellectual humility is crucial for navigating these complexities, enabling academics and practitioners to deliver person-centered and culturally informed care that respects patients’ belief systems, while supporting their possible evolution over time. In doing so, osteopathic care can further advance its commitment to address the full spectrum of human experiences in a meaningful and inclusive manner.

## Figures and Tables

**Figure 1 healthcare-13-00586-f001:**
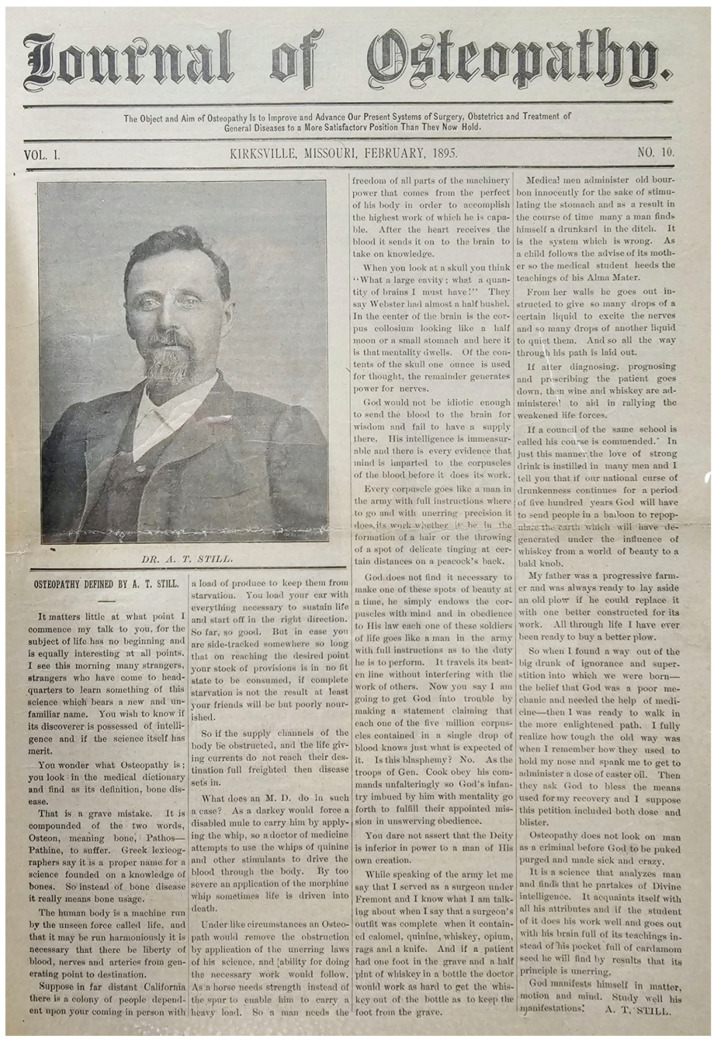
Front page of the Journal of Osteopathy, Vol. One Number 10 February 1895, featuring Dr. A. T. Still’s quote on mind, matter, and motion. Courtesy of the Museum of Osteopathic Medicine, Kirksville, MO, USA.

**Figure 2 healthcare-13-00586-f002:**
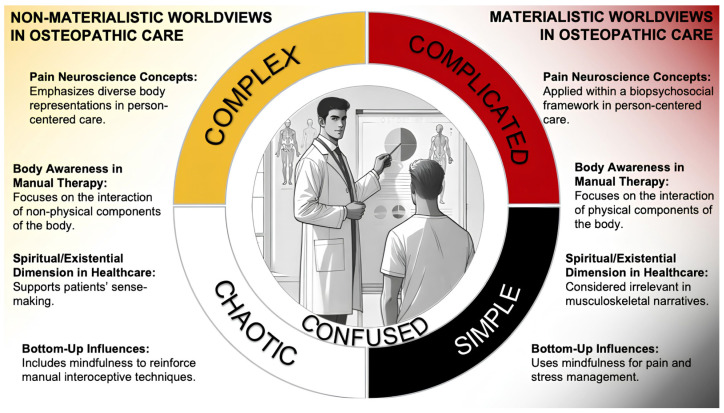
The Cynefin framework integrates materialistic and nonmaterialistic worldviews in osteopathic care. The Cynefin framework (CF) is a decision-making tool that facilitates synchronization between the patient, practitioner, and environment through a narrative-based approach. It identifies four domains and a confused space, highlighting the dynamic relationship between individuals, their experiences, and contexts. The CF promotes epistemological flexibility by integrating materialistic and nonmaterialistic worldviews, helping to balance person-centered osteopathic care with evidence-informed manual therapy. This ensures that care is individualized and aligned with patient values, expectations, and clinical contexts.

**Table 1 healthcare-13-00586-t001:** Comparative framework of Indigenous narratives and body representations versus Western biomedical perspectives.

Level	Indigenous Narratives and Body Representations	Western Biomedical- Focused Perspectives
**Personal Level**	Emphasis on subjective experience, self-awareness, and personal growth	Focus on individual biology and physiology (e.g., genetic makeup, physical health)
	Explores questions of identity, personal history, emotions, values, and purpose	Investigates the brain-body connection, mental health, and individual behavior
	Importance of emotions and internal states in shaping one’s interaction with the world	Psychological and neurological factors are viewed as separate from spiritual or emotional aspects
**Collective Level**	Shared cultural values, roles, and group identity are central	Focus on social biology, social behaviors, and collective systems (e.g., group dynamics, norms)
	Interactions within communities shape identity and the experience of owning a body	Emphasis on societal structures and cultural influences on individual behavior
	Highlights collective rituals, traditions, and intergenerational knowledge	Less emphasis on ritualistic or spiritual practices; more focus on social norms and group behaviors
**Transpersonal Level**	Connection to spiritual, cosmic, or transcendent dimensions of existence	Typically, minimal focus on transcendent or spiritual aspects of the body
	Interconnectedness with all living beings, the environment, and the universe	Focus on the material world, often ignoring spiritual/existential dimensions
	Exploration of universal principles, collective unconscious, and spiritual experiences	Biological determinism and empirical science often exclude spiritual/existential perspectives

This table contrasts Indigenous perspectives on the body, which encompass personal, collective, and transpersonal dimensions, with Western biomedical models that focus primarily on individual biology and material aspects. It outlines how Indigenous worldviews emphasize interconnectedness, spirituality, and community, while Western perspectives prioritize physiological health, social.

**Table 2 healthcare-13-00586-t002:** Managing non-physical components of the body in osteopathic care: avoiding pseudoscience and scientism through critical thinking to align with patients’ values and expectations.

Key Concepts of the Scientific Method Applied to Healthcare	
Intellectual Humility	Recognizing and integrating diverse worldviews to enhance understanding and collaboration
Science as a Bridge	Utilizing scientific methods to connect known and unknown perspectives effectively
Epistemological Flexibility	Acknowledging diverse worldviews and selecting the most suitable epistemological framework to support patients’ sense-making
Theories on Consciousness [6]	
Nonmaterialistic Worldviews	Materialistic Worldviews
Brain as a receptor of consciousness	Brain as an emitter of consciousness
Body as a vehicle for experience (“human being”)	Body as a mechanical system (“human doing”)
The Influence of the Flexner Report on Osteopathic Education [46]	
Nonmaterialistic Worldviews	Materialistic Worldviews
Non-physical components of the body (emotions, existential dimension) considered pseudoscience	Only physical components of the body considered worthy of investigation
Therapeutic framework built around the patient experience (illness model)	Therapeutic framework built around observed biological components (disease model)
Incorporating Indigenous epistemological frameworks in osteopathic care (the body–mind–spirit tenet)	Therapeutic framework built around observed biological components (disease model)
The Cynefin Framework to Guide Culturally Sensitive and Person-Centered Osteopathic Care [7,56]	
Nonmaterialistic Worldviews	Materialistic Worldviews
Pain neuroscience concepts consider the diversity of body representations in person-centered care	Pain neuroscience concepts applied within a biopsychosocial framework in person-centered care
Body awareness in manual therapy focuses on the interaction of non-physical components of the body	Body awareness in manual therapy focuses on the interaction of physical components of the body
The spiritual/existential dimension in healthcare might be included in the narrative to support patients’ sense-making	The spiritual/existential dimension in healthcare is considered not relevant for a musculoskeletal-focused narrative
Bottom-up influences might include mindfulness approaches to reinforce manual interoceptive techniques	Bottom-up influences might include mindfulness approaches for pain management and stress reduction

## Data Availability

Not applicable.

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
