# Peer review of "Indigenous Epistemological Frameworks and Evidence-Informed Approaches to Consciousness and Body Representations in Osteopathic Care: A Call for Academic Engagement"

_healthcare, 2025, doi:10.3390/healthcare13060586_

Round 1
Reviewer 1 Report
Comments and Suggestions for Authors
The authors are to be commended for their desire to bring attention to the value of Indigenous healing traditions for enriching person-centered care for all humans. The paper makes a good statement in summarizing the need to "recogniz[e] that Indigenous knowledge is a living, adaptive knowledge that has much to offer in a world characterized by ecological and health crises." While this goal is good, I think there are a number of necessary ways in which the paper needs to reconsider how it describes the richness and variety of Indigenous healing practices, the specific relationship (both current and historical) between osteopathy and Native American healing traditions, and the precise nature of the desired "bridging" named in the title .
- More specificity in describing Indigenous healing traditions is needed. The paper does an admirable job in introducing and outlining distinctions in Western philosophy and healing practices. It does not offer the same level of specificity in describing distinct Indigenous traditions or figures, which also vary. While I agree with the utility of the broad concepts of "Indigenous" and "Western" ways of knowing, being, and practicing care, the paper reads as unbalanced and having greater knowledge of Western systems. The absence of specific, named traditions and figures within Indigenous practices contributes to this. No specific Native American healing traditions are named or acknowledged, despite the paper citing literature that highlight's A.T. Still's (unacknowledged?) endebtedness to Shawnee, Pawnee, Kickapoo, Cherokee, and Pottawatomie healing traditions. Instead the paper generalizes to: "Native American perspectives significantly influenced Dr. Andrew Taylor Still, whose 19th-century holistic practices continue to shape integrative healthcare." Many other references to general "Indigenous" beliefs, while arguably appropriate, are noticeably not distinguished by more specific references even to being (North) American beliefs or practices.
- In different places the paper says it will "investigate" the Native American "influences on" or "roots" of osteopathy, but in my opinion it does not do this because it does not describe how this influence occurred, which Elders or what cultures A.T. Still learned from, or current day practices in these cultures that continue to influence osteopathy. The paper describes many ways in which there is alignment between person-centered care and Indigenous healing practices, but this is not the same as tracing a lineage. Naming the alignment is also valuable, and perhaps that is a framing that the authors would wish to pursue in revisions, and perhaps position the paper as an exploration of alignment and (mutual?) influence between distinct traditions of equal value. This leads to my third general critique.
- The paper states a desire to "integrat[e] Indigenous narratives and body representations...," as well as to "draw on" these traditions, and in some ways this language of integration and (selective) use troubles me. Something suggests that this integration is being made so as to improve the dominant model in a way that is not recognizing the independent and equal value of Indigenous ways of healing. Indigenous traditions come across as a supplement to a (Western) system of thinking that only needs a bit of tweaking, rather than a radical overhaul, in order to be compatible with Indigenous systems. Indigenous healing traditions then, by extension, are possibly being valued for their utility for Western systems, rather than in their own right; the idea that they might provide foundations for future integrated care systems is not explored. In many places in the paper the stark contrast between these different healing traditions is outlined, and the reductionist limitations of the biomedical model are named. Other readers might disagree, but I feel the "challenges section" is too vague on naming these underlying challenges, including the role of history in shaping care systems which has led to the ongoing "marginalization" of Indigenous belief systems, or in acknowledging the possible dangers of appropriation of knowledge without appropriate training or context.
- A minor point: I had difficulty understanding some aspects/purpose of Table 2, in particular heading references to the Flexner report and the Cynefin model, which are not discussed in the text.
Reviewer 2 Report
Comments and Suggestions for Authors
ARTICLE REVIEW
The article addresses an important and much-needed topic: introducing diverse paradigms of the body and health by integrating various philosophical and medical approaches in today's globalized world. Our era is characterized by medical pluralism and the growing emphasis on patient-centered care within healthcare systems. The topic has been relevant since the early days of medical anthropology, which challenged Western biomedical paradigms, and its importance has only increased in the context of decolonization and the urgent need to include diverse epistemologies in contemporary discourse. A notable strength of the article lies in its inclusion of neuroscience and theories of consciousness, which enrich the discussion of the connection between body, mind, and spirit (as the article proposes). These perspectives provide an opportunity to deepen our understanding of health and healing.
However, while the topic is highly relevant and offers significant potential for analysis, the paper falls short of fully addressing its objectives. For instance, the connection between osteopathy and North American Indigenous healing traditions (which ones?) remains vague and underdeveloped. Although the article frequently references this relationship, the only evidence provided is the claim that the founder of osteopathy had contact with Indigenous North American populations. This alone does not substantiate the assertion that Indigenous traditions directly inspired osteopathic principles.
Additionally, the introduction includes repetitive arguments that Indigenous medicine could enhance contemporary paradigms of health and the body. This could be the case, of course, but the article never explains how that could take place. The article subsequently relies on foundational theories from medical anthropology—such as dominance of Cartesian dualism in Western societes, Kleinman’s distinction between disease and illness, and Scheper-Hughes and Lock’s tripartite body framework. While these theories have shaped the discipline, they are now part of much-discussed classical literature. Their inclusion without engaging with more recent and nuanced perspectives limits the paper’s analytical depth and contemporary relevance.
In its conclusion, the article moves toward theories of consciousness and promises to address connections between brain activity and lived experiences. Yet, this analysis remains superficial. Similarly, the remark on practitioners with "epistemological flexibility," who could pioneer new approaches to the body-mind connection and self-healing, is intriguing but underexplored.
Moreover, the methodology section is entirely absent, which is a significant gap. Without this section, it is difficult to evaluate the research framework or assess the rigor of the paper’s claims.
Finally, the article would benefit from a more critical approach to Indigenous knowledge. The author is encouraged to avoid romanticization and compartmentalization, as these tendencies risk tokenizing and commodifying Indigenous practices rather than fostering shared epistemologies that empower these traditions. Incorporating this perspective and addressing the outlined weaknesses could make the article a strong candidate for publication.
Reviewer 3 Report
Comments and Suggestions for Authors
Dear Authors,
Thank you for submitting your manuscript. While your work explores an important and timely topic, there are several significant issues that, unfortunately, prevent this paper from being suitable for publication in its current form. Below, I have outlined the primary concerns:
1. The manuscript relies disproportionately on self-citations, which raises questions about the diversity and breadth of perspectives presented. Incorporating a wider range of sources would strengthen the argument and demonstrate engagement with the broader field. Additionally, the manuscript does not consistently adhere to MDPI’s citation style, with incomplete references, mismatched formats, and deviations from prescribed guidelines. Ensuring proper citation practices is critical for maintaining scholarly rigor and you should try to implement this practice.
2. The quality of English throughout the manuscript is suboptimal. Many sentences are verbose, grammatically incorrect, or awkwardly phrased, which diminishes clarity and professionalism. Furthermore, key terms and concepts are insufficiently defined, and redundant language weakens the overall narrative. A thorough revision for grammar, style, and conciseness is necessary to enhance readability. Actually, the manuscript lack of a correct flow.
3. The paper lacks a focused research question or hypothesis, resulting in a disjointed narrative that obscures its intended contribution to the field. Without a clear objective, it is challenging to evaluate the significance of the work or its alignment with the scope of the journal. This is one of the most important issue of your work. I suggest you to improve this aspect.
4. While the manuscript engages with theoretical frameworks, it does not provide actionable strategies or concrete examples for integrating Indigenous perspectives into osteopathic care. Practical implications are essential for bridging the gap between theory and clinical practice, and their absence significantly limits the paper's impact.
5. The manuscript places a strong emphasis on non-materialist theories, but it does so without sufficient empirical evidence or critical engagement with alternative perspectives. This undermines the credibility of the arguments presented and risks the work being perceived as pseudoscientific. A balanced discussion, supported by robust evidence, is necessary to establish the legitimacy of these concepts. According to this, I encourage a deep review of this work.
6. The manuscript repeats several ideas, particularly concerning the historical context and holistic principles of osteopathy, which detracts from the overall flow and coherence. Consolidating these discussions and focusing on novel insights would improve the structure of the paper. This aspect can be addressed after implementing the suggested revisions to the earlier points.
Comments on the Quality of English LanguageSee my main report and suggestions about the use of language, grammar, and form.
Round 2
Reviewer 1 Report
Comments and Suggestions for Authors
Thank you to the authors for their detailed response and the thoughtful edits in the paper. It was helpful to hear in the response letter more of the motivation and context, and to have it pointed out that this paper is in response to a special issue call, which helps make sense of why the authors chose their particular framing. The writing is improved in many small but cumulatively impactful ways.
There are some ways in which I think our perspectives simply differ, and in the spirit of intellectual humility which is one of the paper’s main calls, I think we can accept that we have different concerns and focal points. The authors, I believe, are concerned to avoid stereotypes or the suggestion that only Indigenous people can benefit from Indigenous wisdom, which is laudable and important. I am more concerned that 1)the richness, depth, and variation of Indigenous knowledge as independent healing traditions and body belief systems may be glossed over, or 2) that, without nuance, advocating universalist benefits may lead readers to value this knowledge primarily as a supplement to Western knowledge and not in its own right. The paper has made some changes in language that help diminish this concern. In my opinion, it could still do more, but I understand the authors have their eyes on a different goal.
I have one lengthy comment/suggestion, and then a couple short ones.
- In my initial review, I suggested an imbalance between the way in which the paper names Western healing traditions and literature more specifically than Indigenous traditions. The authors have perhaps misunderstood my critique, in that they responded by saying they did not want to limit the applicability of the paper to a specifically Native American context. I am entirely in agreement that Indigenous healing traditions and approaches to the body have much value for contributing to a universalist, person-centered care and take no issue with this. While I think the edits in response were improvements that make the intent of the paper’s global context clearer, I still see the imbalance in that Western knowledge is described with greater specificity. I acknowledge the authors are in a tricky, maybe impossible spot, since Indigenous cultures often have a different orientation to knowledge creation which makes the concept of authorship challenging and which often makes fewer claims to the universality of knowledge out of context; at the same time, Indigenous research is often very explicit about naming whose knowledge is being cited, and Indigenous researchers do seek to have their work published and acknowledged. I would encourage the authors to consider two things. 1) Name this paradox of the desire to give credit to specific Indigenous healers and thinkers, but recognizing both that the marginalization of this knowledge means it is less often cited in the literature, as well as the context that Indigenous cultures often have different values around who “owns” and can use knowledge that make Western citation and acknowledgement difficult. 2) I count approximately 10 times where, in describing characteristics or practices of Western healthcare or science , the authors who wrote the paper or originated the idea are named in the sentence (not just cited with a number). In contrast, there are no instances, I believe, where descriptions of Indigenous traditions and beliefs name the authors in the sentence. This has the effect of burying the possibility that Indigenous healthcare and science is equally populated by different strands of thought, philosophy, and healing approaches. As an aside, the issue here is not whether the authors are Indigenous, though it’s a separate one worth considering, but rather that, in this subtle difference, the paper suggests that our definition of Indigenous healing or thought is less shaped by who is writing about it. This could inadvertently contribute to a stereotype of Indigenous practices as timeless and impersonal, which is exactly what the authors have stated they wish to avoid. This could be easily changed and would help address the concern that Indigenous traditions being treated as more monolithic and/or that our understanding of what they are is independent of who is writing and thinking about them. I would recommend naming individual writers about health and science in similar ways when discussing what “Western” and “Indigenous” science and or health are.
Overall, the edits to the paper improve clarity. Table 2 is improved but the last section on pseudo-science and scientism still feels like it doesn’t hang together with the rest of the table, and the heading of “the legacy and implication” doesn’t clarify things. I would drop this last section from the table or make it a separate table of its own.
There is a sentence which says, “may challenge academics and conventional Western epistemologies,” I would suggest changing this to “may challenge conventional Western academics and epistemologies” – this small change would acknowledge that not all academics are (or must be) rooted in conventional Western epistemologies.
There is a sentence which says, “This flexibility acknowledges the historical connections between Dr. A.T. Still and Native American populations” consider whether “Native American healers” would be more appropriate. To me, “populations” implies Dr. Still learned by indirect observation of what Native healers practiced, while “healers” implies he was in dialogue with or was directly taught by specific teachers. I do not know the history to know which the authors feel is more accurate.
There is a sentence which suggests that the goal is “facilitating the integration of these perspectives with conventional Western reductionist approaches to provide care tailored to patients' specific values and expectations in clinical care.” I feel that successful integration will require some transformation of reductionist approaches and think the authors might find slightly better wording to convey this.
At line 389, for consistency with the rest of the paragraph, it should be “conceptualize” not “conceptualized.”
The new title more accurately conveys the authors’ aims and the content of the paper. However I would suggest returning to “Indigenous traditions AND (rather than ‘to’) evidence-informed approaches.” The “to” implies a spectrum running from one to the other, and while there is a kind of “evidence” that is narrow (and under critique) in the paper, the paper advocates for a broader concept of evidence, so I would not set the concept of evidence up in opposition to Indigenous tradition.
Reviewer 2 Report
Comments and Suggestions for Authors
The article addresses a highly relevant and timely topic: the introduction of diverse paradigms of the body and health by integrating various philosophical and medical approaches in today’s globalized world. The topic has been central to medical anthropology since its inception, challenging Western biomedical paradigms, and has only grown more critical in the context of decolonization. A significant strength of the article is its inclusion of neuroscience and theories of consciousness.
The revised version of the article has explained, in more necessary detail, the connection between osteopathy and North American Indigenous healing traditions, offering more developed insights that provide clearer evidence of how these traditions have directly influenced osteopathic principles. The relationship is now better supported by thoughtful references and examples that substantiate the claim, strengthening the paper’s argument.
Furthermore, the introduction thoughtfully explores the potential of Indigenous medicine to enhance contemporary paradigms of health and the body, providing a clear explanation of how this integration could take place. The article also more successfully engages with more contemporary perspectives, building on foundational medical anthropology theories. By incorporating recent and nuanced developments in the field, the paper enhances its analytical depth and contemporary relevance.
In its conclusion, the article moves adeptly toward theories of consciousness, offering a rich and comprehensive analysis of the connections between brain activity and lived experiences. The discussion surrounding practitioners with "epistemological flexibility" is also now better explored, providing an exciting vision for how these individuals could pioneer new approaches to the body-mind connection and self-healing.
The methodology section has been thoroughly addressed, offering a clear and robust framework that strengthens the paper’s rigor and allows for a comprehensive evaluation of its claims. It should be included in its integral form.
Finally, the article now demonstrates a critical and respectful approach to Indigenous knowledge. The author avoids romanticization and compartmentalization, instead fostering shared epistemologies. This nuanced and thoughtful approach greatly enhances the overall quality of the paper, making it a better candidate for publication.
Reviewer 3 Report
Comments and Suggestions for Authors
Dear Authors,
Thank you for resubmitting your manuscript. I appreciate the time and effort you dedicated to revising your work.
After careful evaluation of your revised submission, I regret to inform you that your manuscript was not revised according to the reviewer's suggestions. While I recognize your efforts to address the comments, significant concerns raised remain unresolved.
Specifically, the manuscript:
- Continues to rely heavily on self-citations, which limits engagement with broader perspectives in the field.
- Has unresolved issues with clarity, flow, and coherence, despite some language improvements.
- Lacks a clear and focused research question or hypothesis, leaving its contribution to the field ambiguous.
- Does not provide sufficient actionable strategies or concrete examples for applying the theoretical concepts discussed.
- Presents a strong emphasis on non-materialist theories without adequate empirical evidence or balanced engagement with alternative perspectives, which undermines its credibility.
While your revisions addressed certain aspects of the reviewers’ feedback, a more thorough reworking of the manuscript would be required to meet the standards of Healthcare. I encourage you to carefully consider all the comments and refine your work to strengthen its clarity, focus, and scholarly rigor.
